# Disorder of Coagulation-Fibrinolysis System: An Emerging Toxicity of Anti-PD-1/PD-L1 Monoclonal Antibodies

**DOI:** 10.3390/jcm8060762

**Published:** 2019-05-29

**Authors:** Ryo Sato, Kosuke Imamura, Shinya Sakata, Tokunori Ikeda, Yuko Horio, Shinji Iyama, Kimitaka Akaike, Shohei Hamada, Takayuki Jodai, Kei Nakashima, Shiho Ishizuka, Nahoko Sato, Koichi Saruwatari, Sho Saeki, Yusuke Tomita, Takuro Sakagami

**Affiliations:** 1Department of Respiratory Medicine, Graduate School of Medical Sciences, Kumamoto University, Honjo 1-1-1, Chuo-ku, Kumamoto-shi, Kumamoto 860–8556, Japan; ryosato.1981@gmail.com (R.S.); imakou1013@gmail.com (K.I.); sakata-1027@hotmail.co.jp (S.S.); yu1980327@gmail.com (Y.H.); iyama.shinji@kuh.kumamoto-u.ac.jp (S.I.); demio0601@gmail.com (K.A.); unagicurry@yahoo.co.jp (S.H.); jojojojojody@gmail.com (T.J.); keinakasima056@gmail.com (K.N.); zhi4_sui4@yahoo.co.jp (S.I.); nahoko.t.k.d@gmail.com (N.S.); ksaruwat@kuh.kumamoto-u.ac.jp (K.S.); saeshow@kuh.kumamoto-u.ac.jp (S.S.); stakuro@kumamoto-u.ac.jp (T.S.); 2Department of Clinical Investigation, Kumamoto University Hospital, Honjo 1-1-1, Chuo-ku, Kumamoto-shi, Kumamoto 860–8556, Japan; ryousei@kumamoto-u.ac.jp

**Keywords:** immune-related adverse events (irAEs), immune checkpoint inhibitor (ICI), non-small-cell lung cancer (NSCLC), programmed cell death 1 (PD-1), programmed cell death ligand 1 (PD-L1), T cell, tissue factor (TF), Trousseau’s syndrome, monocyte, macrophage

## Abstract

A disruption of immune checkpoints leads to imbalances in immune homeostasis, resulting in immune-related adverse events. Recent case studies have suggested the association between immune checkpoint inhibitors (ICIs) and the disorders of the coagulation-fibrinolysis system, implying that systemic immune activation may impact a balance between clotting and bleeding. However, little is known about the association of coagulation-fibrinolysis system disorder with the efficacy of ICIs. We retrospectively evaluated 83 lung cancer patients who received ICI at Kumamoto University Hospital. The association between clinical outcome and diseases associated with disorders of the coagulation-fibrinolysis system was assessed along with tumor PD-L1 expression. Among 83 NSCLC patients, total 10 patients (12%) developed diseases associated with the disorder of coagulation-fibrinolysis system. We found that disorders of the coagulation-fibrinolysis system occurred in patients with high PD-L1 expression and in the early period of ICI initiation. In addition, high tumor responses (72%) were observed, including two complete responses among these patients. Furthermore, we demonstrate T-cell activation strongly induces production of a primary initiator of coagulation, tissue factor in peripheral PD-L1^high^ monocytes, in vitro. This study suggests a previously unrecognized pivotal role for immune activation in triggering disorders of the coagulation-fibrinolysis system in cancer patients during treatment with ICI.

## 1. Introduction

T cell activation and proliferation are initiated through antigen recognition by the T cell receptor (TCR). The T cell response is regulated by a balance between co-stimulatory and inhibitory signals called immune checkpoints [1]. Under normal physiological conditions, immune checkpoints play a crucial role in maintaining immune homeostasis and preventing autoimmunity [1]. In cancer, immune checkpoint pathways delivering inhibitory signals are often activated to suppress anti-tumor immune response in tumor immune microenvironments as one of the mechanisms of tumor immune escape [2,3,4,5].

The first generation of antibody-based immunotherapy, called immune checkpoint inhibitors (ICIs), blocks the receptor and/or ligand interactions of molecules, such as cytotoxic T-lymphocyte antigen 4 (CTLA-4), programmed cell death 1 (PD-1), and programmed cell death ligand 1 (PD-L1) [1,5]. Anti-PD-1/PD-L1 antibodies inhibit the interaction between PD-1 and PD-L1 and unleash immune responses against tumors: activating or boosting the activation of the immune system to attack cancer cells [6]. The immunotherapy targeting the PD-1/PD-L1 pathway has shown significant and durable clinical responses for non-small-cell lung cancer (NSCLC) patients in addition to a more favorable toxicity profile and improved tolerability than chemotherapy [7]. Anti-PD-1/PD-L1 antibody therapy has changed the treatment landscape and led to a paradigm shift in treatment strategies in NSCLC, and is now a standard-of-care for NSCLC [6,7,8]. ICI are recognized as a promising strategy to treat various types of cancer and the indications for the use of ICIs continue to expand with unprecedented speed [5,6,7,8]. 

A disruption of inhibitory immune checkpoint leads to imbalances in immune homeostasis, resulting in adverse effects which are termed immune-related adverse events (irAEs) that share clinical features with autoimmune diseases or inflammatory diseases [9]. The irAEs can affect multiple organs of the body and are commonly seen in the skin, lungs, thyroid, endocrine, adrenal, pituitary, gastrointestinal tract, musculoskeletal, renal, and nervous system [9,10]. Most of the irAEs are usually reversible, but in rare cases, they can be severe and life-threatening. In addition, unexpected severe irAEs have emerged in real-world clinical practice [11,12,13,14,15]. Thus, elucidating mechanisms of irAEs is urgently needed to improve their early diagnosis and develop more precise treatments for irAEs [9,10]. 

A balance between clotting and bleeding is maintained in normal physiology, but can be altered under the presence of malignancies [16,17,18]. It has been known that a coagulation homeostasis could be further impaired after nonsurgical cancer therapy including radiation therapy, standard chemotherapy, and targeted therapy, which could trigger both bleeding and thrombosis; however, the underlying mechanisms remain unclear [18,19,20]. Recently, several case studies have suggested that anti-PD-1/PD-L1 monoclonal antibodies might trigger disorders of the coagulation-fibrinolysis system in advanced cancer patients, which implies that the systemic immune activation may impact a balance between clotting and bleeding [13,14,21,22,23]. However, the association of coagulation-fibrinolysis system disorder with the efficacy of anti-PD-1/PD-L1 monoclonal antibody therapies and clinical characteristics of the patients who develop diseases associated with disorders of the coagulation-fibrinolysis system under treatment with ICIs have not been studied yet.

Tissue factor (TF) is a transmembrane cell surface glycoprotein that triggers the extrinsic coagulation cascade and is essential for hemostasis. TF binds the coagulation serine protease factor VII/VIIa (FVII/VIIa) to form a bimolecular complex that functions as the primary initiator of coagulation in vivo [24,25]. Studies have shown that levels of circulating TF in the form of microparticles are increased in various diseases, including cardiovascular disease, sepsis, and cancer [16,26]. In addition, circulating TF in blood has been suggested to be a cause of distant thromboses and contributes to the increased incidence of thrombosis observed in these diseases. Importantly, monocytes have been shown to be the major source of intravascular TF in many diseases [16,25,27]. Therefore, we focused on the relationship between T cell activation and induction of TF expression on monocytes in peripheral blood mononuclear cells (PBMCs) in this study. 

Here, we report the clinical features of NSCLC patients who developed diseases associated with disorders of the coagulation-fibrinolysis system during treatment with ICI. We demonstrate that T cell activation leads to promoting production of a primary initiator of coagulation, TF, in PD-L1^high^ human peripheral CD14^+^ monocytes. We also discuss the underlying mechanisms of the onset of disorders of coagulation-fibrinolysis system as an irAE of anti-PD-1/PD-L1 antibody therapy. This study suggests a previously unrecognized pivotal role for immune activation in triggering disorders of coagulation-fibrinolysis system in advanced cancer patients during treatment with ICI monotherapy.

## 2. Materials and Methods

### 2.1. Patients

The medical records of patients with advanced NSCLC who had received nivolumab (3 mg/kg every 2 weeks), pembrolizumab (200 mg every 3 weeks), or atezolizumab (1200 mg every 3 weeks) monotherapy at Kumamoto University Hospital between January 2016 and October 2018 were retrospectively reviewed. Treatments were provided until disease progression, unacceptable toxicity, or consent withdrawal. The present study was approved by the Kumamoto University Institutional Review Board (IRB number, 1685, Approval Date, 27 March 2018.)

To maximally characterize the clinical features of NSCLC patients who developed diseases associated with disorders of the coagulation-fibrinolysis system during immune checkpoint blockade therapy, we searched the PubMed database. We did not limit search dates and we looked for articles published in English.

### 2.2. Assessments

Only adverse events associated with disorders of the coagulation-fibrinolysis system (thromboembolic and bleeding complications), which were not detected before treatment with ICI and newly developed during treatments with ICI (within 30 days of the last administration of ICI), were considered as disorders of the coagulation-fibrinolysis system possibly triggered by immune checkpoint blockade. The clinical severity of coagulation-fibrinolysis system disorders was graded according to the Common Terminology Criteria for Adverse Events, version 5.0. A newly developed purpura involving more than 10% of the body surface area (grade ≥ 2) was assessed as one of the bleeding complications. Disorders of the coagulation-fibrinolysis system accompanied with the abnormal decrease in platelet count less than 100 × 10^9^/L were not considered as coagulation-fibrinolysis system disorders triggered by ICI to exclude immune thrombocytopenia which previously reported as a rare irAE [17,28]. Tumor response to nivolumab, pembrolizumab, or atezolizumab monotherapy was objectively assessed by pulmonary physicians according to Response Evaluation Criteria in Solid Tumors, version 1.1. The Kaplan–Meier method was used to obtain estimates of progression-free survival (PFS) and overall survival (OS). PFS was measured from the date ICIs started to the date of documented progression or death. Patients who were alive and not known to have progressed were censored. OS was measured from the date ICI started to the date of death or last follow-up. The data cutoff date was 15 March 2019. 95% confidence intervals of survival were calculated by the log–log transformation of survival. The analysis was performed using GraphPad Prism 7.0c software (GraphPad Software, San Diego, CA, USA). 

### 2.3. PD-L1 Staining

PD-L1 expression in the lung cancer specimen was analyzed by immunohistochemical staining using the PD-L1 IHC 22C3 pharmDx antibody (clone 22C3, Dako North America, Inc., Carpinteria, CA, USA). The antibody was applied according to DAKO-recommended detection methods. PD-L1 expression in tumor cells was scored as the percentage of stained cells.

### 2.4. Isolation of PBMCs

Blood samples of healthy donors were collected in cell preparation tubes with sodium citrate (BD Vacutainer CPT Tubes, BD Biosciences, Franklin Lakes, NJ, USA). PBMCs were obtained by centrifugation following the manufacturer’s protocol.

### 2.5. In Vitro Assay

Isolated PBMCs of healthy donors (2 × 10^6^/well) were activated with lipopolysaccharide (LPS) (100 ng/mL; *Escherichia coli* 0111:B4, Sigma-Aldrich, St. Louis, MO, USA) or CD3/CD28/CD2 beads (T-cell Activation/Expansion Kit, Miltenyi Biotec, Bergisch Gladbach, Germany) for 18 h in 24-well flat bottom plates with 2 mL RPMI 1640 medium (FUJIFILM Wako Pure Chemical Corporation, Osaka, Japan) supplemented with 10% fetal bovine serum (Biological Industries, Kibbutz Beit Haemek, Israel) at 37 °C and 5% CO2. After 18 h, flow cytometric analysis and immunofluorescence staining were performed.

### 2.6. Flow Cytometric Analyses

Multiparameter flow cytometric analysis was performed on PBMCs. Briefly, cells were incubated with Fc receptor blocking agent (Miltenyi Biotec, Bergisch Gladbach, Germany) and stained with monoclonal antibodies for 20 min at 4 °C in a darkened room. CD3 and CD14 immunophenotypic markers were used to define T lymphocytes and monocytes. Each population was also evaluated for CD142 (tissue factor; TF), and PD-L1 expression. The following monoclonal antibodies were used (all from BioLegend, San Diego, CA, USA): FITC-CD3 clone OKT3, PerCP/Cy5.5-CD14 clone HCD14, APC-CD69 clone FN50, PE-CD142 clone NY2, PE/Cy7-HLA-DR clone L243, Brilliant Violet 421-PD-L1 clone 29E.2A3 were used. Matched isotype controls were used for each antibody to establish the gates. Live cells were discriminated by means of LIVE/DEAD Fixable Aqua Dead Cell Stain (Thermo Fisher Scientific, Waltham, MA, USA) and dead cells were excluded from all analyses. All flow cytometric analyses were performed using a BD FACSVerse™ (BD, Franklin Lakes, NJ, USA). Data were analyzed using FlowJo software (FlowJo LLC, Ashland, OR, USA).

### 2.7. Immunofluorescence Staining

Immunofluorescence staining was performed on PBMCs. Briefly, cells were incubated with Fc receptor blocking agent (Miltenyi Biotec, Bergisch Gladbach, Germany) and stained with monoclonal antibodies for 20 min at 4 °C in a darkened room. The following monoclonal antibodies were used (all from BioLegend, San Diego, CA, USA): FITC-CD3 clone OKT3, PerCP/Cy5.5-CD14 clone HCD14, APC-CD69 clone FN50, PE-CD142 clone NY2. After fixation of stained cells using Fix/Perm buffer (Thermo Fisher Scientific, Waltham, MA, USA), the suspension of fixed cells was immobilized onto glass slides by cytospin. Nuclei were counter stained with 4’,6-diamidino-2-phenylindole dihydrochloride (DAPI) (DOJINDO, Kumamoto, Japan) in water, and whole sections were mounted in ProLong Diamond (Thermo Fisher Scientific, Waltham, MA, USA). Slides were observed with a confocal fluorescence microscope (FV3000, Olympus, Tokyo, Japan).

## 3. Results

### 3.1. Disorder of Coagulation-Fibrinolysis System Triggered by Immune Checkpoint Blockade in Advanced Lung Cancer

Only diseases associated with disorders of coagulation-fibrinolysis system occurred during treatment with ICI were considered as the possible irAEs triggered by immune checkpoint blockade [13,14,21,22,29]. Disorders of the coagulation-fibrinolysis system accompanied with the abnormal decrease in platelet count were not considered as coagulation-fibrinolysis system disorders triggered by ICI to exclude immune thrombocytopenia which previously reported as a rare irAE [28]. Disseminated intravascular coagulation (DIC) caused by pneumonia and sepsis accompanied with elevations of procalcitonin in blood were seen in two patients during treatment with ICI. However, ICI-related DIC without infectious diseases was not observed in current study. Thus, the two patients who developed DIC were not considered as having coagulation-fibrinolysis system disorders triggered by ICI.

Among 83 advanced NSCLC patients receiving nivolumab, pembrolizumab, or atezolizumab monotherapy at Kumamoto University Hospital between January 2016 and October 2018, a total of 10 patients (12%) developed diseases associated with the disorder of coagulation-fibrinolysis system (thromboembolic and bleeding complications) during treatment with ICI, of which 2 patients were cases recently reported from our group [13,14]. To maximally characterize the clinical features of NSCLC patients who developed diseases associated with disorders of the coagulation-fibrinolysis system during immune checkpoint blockade therapy, we added two NSCLC cases identified by the PubMed database search to this study [22,29]. Ferreira et al. have shown a case of NSCLC patient who had an acute coronary syndrome (ACS) during the second administration of nivolumab (Table 1 and Table 2) [22]. Kunimasa et al. have reported a case of deep vein thrombosis (DVT) and pulmonary thromboembolism (PTE) associated with pembrolizumab in NSCLC patient (Table 1 and Table 2) [29]. The characteristics of a total 12 patients who developed diseases associated with disorder of the coagulation-fibrinolysis system are summarized in Table 1 and Table 2.

The median age was 70.5 (range, 48–81) years. Of 12 patients, four (33%) were diagnosed as having squamous cell carcinoma and eight (67%) were diagnosed as having nonsquamous NSCLC. Mutation (L858R) in *epidermal growth factor receptor* (*EGFR*) was present in one patient (8%). Six patients (50%) had undergone no prior chemotherapy regimen, whereas 6 of 12 patients (50%) had received one or more previous chemotherapies. The median treatment line was 1.5 (range, 1–4). 

A broad spectrum of diseases associated with disorders of coagulation-fibrinolysis system including ACS (*n* = 2), cerebral infarcts (CI; *n* = 5; Figure 1, case 4), DVT (*n* = 3; Figure 1, case 9), PTE (*n* = 1), intra brain tumor hemorrhage (*n* = 1), gastrointestinal bleeding (*n* = 2), purpura involving more than 10% of the body surface area (*n* = 2), and bronchial hemorrhage (*n* = 2) associated with the administrations of ICIs was observed (Table 1). Of the 12 patients, 11 were treated with anti-PD-1 monoclonal antibodies (nivolumab or pembrolizmab), while 1 patient received an anti-PD-L1 monoclonal antibody (atezolizumab). The severe grade (≥Grade 3) of diseases associated with disorders of coagulation-fibrinolysis system were seen in 4 out of 12 patients (33%). One death related to intra brain-tumor hemorrhage was observed. The first onset of diseases associated with disorder of coagulation-fibrinolysis system developed within 2 cycles of anti-PD-1/PD-L1 monoclonal antibody therapies for 8 patients (67%), with a median onset of 1 cycle (range 1–16 cycles) (Table 1). Among 12 patients, 4 patients had past medical history of diseases associated with disorders of coagulation-fibrinolysis system. Three of the 4 patients had been taking antiplatelet agents (Table 2). Antiangiogenic agents including bevacizumab have not been used prior to ICI in these 12 patients.

### 3.2. Association of Coagulation-Fibrinolysis System Disorders with the Efficacy of Anti-PD-1/PD-L1 Monoclonal Antibody Therapies in NSCLC

The PD-L1 tumor proportion score (TPS) were evaluable in 11 of 12 patients who developed diseases associated with disorders of the coagulation-fibrinolysis system (Table 1). Interestingly, all 11 patients were positive for PD-L1. The expression of PD-L1 was abundant (TPS ≥ 50%) in 9 of 11 patients (82%) (Figure 2A) and at low levels (1% ≤ TPS < 50%) in 2 patients (18%). The association between irAEs and the efficacy of anti-PD-1/PD-L1 monoclonal antibody therapies in NSCLC have been reported [5,30,31,32,33]. Of the 12 NSCLC patients, 11 were assessable for tumor response. One patient died 28 days after starting treatment due to pneumonia and was not assessable for response. Based on Response Evaluation Criteria in Solid Tumors, version 1.1, complete response (CR) was observed in two patients (17%), partial response in six patients (55%), stable disease in two patients (18%), and progressive disease in one patient (9%). The objective response rate (ORR) and the disease control rate (DCR) were 72% and 91%, respectively in NSCLC patients evaluated for tumor response (*n* = 11) (Figure 2B). Of 12 NSCLC patients who developed diseases associated with disorders of the coagulation-fibrinolysis system, 10 were assessable for survival. The 10 patients analyzed for PFS and OS had received ICIs at Kumamoto University Hospital. The median PFS was 8.3 months (Figure 2C, left panel). The median OS was not reached (Figure 2C, right panel).

Chemotherapy and radiotherapy in patients with advanced cancer have been known to not only trigger coagulation disorders, but also enhance the risk of bleeding complications due to the local fibrinogen and platelet consumption and induction of endothelial injury [16,17,18,19,20,34]. In the current study, 7 out of 12 patients (58%) had hemorrhagic complications after immune checkpoint blockade therapy (Table 1). Case 10, 11, and 12 showed only hemorrhagic complications (bronchial hemorrhage, gastrointestinal bleeding, and grade 2 purpura). Among hemorrhagic complications, hemorrhage from brain tumor (case 2, Table 1) was a grade 5 adverse event, suggesting clinicians should be aware of the risk of hemorrhagic complications during ICI therapy as a potential life-threatening irAE [13].

### 3.3. T Cell Activation Induce Tissue Factor Expression on PD-L1-Positive Monocytes

LPS-activated monocytes have been known to produce abundant TF [17,35,36]. Consistent with the results reported previously, TF was expressed on LPS-activated CD14^+^ monocytes (Figure 3A,B). However, activated T cells did not express TF (Figure 3A). 

Little is known about the impact of T cell activation on TF production in human circulating PD-L1 expressing monocytes. Thus, we next assessed the impact of activated T cells on TF expression on monocyte in vitro. To activate T cells, PBMCs from healthy donors were incubated with CD3/CD28/CD2 beads, which can provide physiological activation of human T cells. T cell activation was confirmed by the surface expression of CD69 and HLA-DR on CD3^+^ T cells. T cell activation by CD3/CD28/CD2 beads induced significant TF expression on CD14^+^ monocytes, which was confirmed by multiparameter flow cytometric analysis and multicolor immune immunofluorescence staining (Figure 3A,B). Activated monocytes have been shown to express PD-L1 and HLA-DR on cell surface, and the PD-L1 suppresses tumor-specific T cell immunity in physiological condition [37,38]. T cell activation by CD3/CD28/CD2 beads markedly increased PD-L1 on monocytes (Figure 4A) and HLA-DR, suggesting activated T cells induced monocyte activation. Interestingly, PD-L1^high^ monocytes expressed higher TF compared to PD-L1^low^ monocytes (Figure 4B). These results suggest that T cell activation leads to monocyte activation and induce high TF expression on peripheral PD-L1^+^ monocytes.

Taken together, our data suggest that although, in physiological condition, upregulated PD-L1 on activated antigen-presenting cells (APCs) suppresses the activated T cells and results in end of immune activation as a homeostatic mechanism, T cell activation by ICIs has the potential to induce abundant TF production on APCs and may trigger disorders of the coagulation-fibrinolysis system (Figure 5) [16,17,25,36,39]. 

## 4. Discussion

A complex interplay between anti-PD-1/PD-L1 monoclonal antibody therapy and host immunity leads to unleash the antitumor immune response, however, the disruption of immune checkpoint signaling also leads to imbalances in immunologic tolerance resulting in an unfavorable immune response which clinically manifest as irAEs [9,10,11,12,13,14]. The number of indications for use of ICIs are growing at an unprecedented speed and ICIs have changed the clinical practice, whereas unexpected irAEs have emerged in the real-world clinical practice [9,10,12,13,14].

In current study, we showed 12 % of NSCLC patients receiving ICI monotherapies developed diseases associated with disorders of the coagulation-fibrinolysis system. 

Interestingly, we found that disorders of the coagulation-fibrinolysis system occurred in patients with high PD-L1 expression on tumor cells and in the early period of ICI initiation. In addition, high tumor responses were observed including two CR cases among these patients, suggesting an association between immune activation by ICIs and the onset of disorders of the coagulation-fibrinolysis system (Table 1). 

The coagulation-fibrinolysis system disorders have not been reported as irAEs in landmark clinical trials of ICIs in cancer patients. In most phase II/III clinical studies of ICIs, only AEs which were considered by the investigators to be related to the study therapy or high-incidence AEs (≥5–10% of patients who received ICI) have been reported (Table 3) [33,40,41,42,43]. Thus, not all AEs were known. AEs associated with disorders of the coagulation-fibrinolysis system have not been reported in five landmark clinical studies of ICIs in patients with advanced NSCLC, whereas hemoptysis (*n* = 16, 6%), pulmonary embolism (*n* = 1, <1%) and cerebrovascular accident (*n* = 1, <1%) were reported in CheckMate057 [44]. However, CheckMate057 have not shown that the association of coagulation-fibrinolysis system disorder with the efficacy of ICIs.

Recently, several case studies have shown that the relationship between the administration of ICIs and the occurrence of diseases associated with disorders of the coagulation-fibrinolysis system, in addition, some of the cases were severe and life-threatening, suggesting ICIs might impact coagulation-fibrinolysis system in advanced cancer patients [13,14,21,22,23,29]. Understanding the underling mechanisms of disorders of coagulation-fibrinolysis system caused by ICIs and clinical characteristics of the patients who develop them are urgently needed to improve their early diagnosis and develop more precise treatments for the adverse events.

### 4.1. Link between PD-L1 Expression on Tumor Cells and Efficacy of ICIs and Disorders of Coagulation-Fibrinolysis System Triggered by ICIs

In this study, we investigated the clinical characteristics of the patients who developed diseases associated with disorders of coagulation-fibrinolysis system under treatment with ICI monotherapy. Although 5 of 18 adverse events (27%; Table 1) associated with disorders of coagulation-fibrinolysis system were severe (Grade ≥ 3), the ORR was 72%. Furthermore, more than 90% of patients with advanced NSCLC achieved disease control. This is the first study to show benefit for patients with advanced NSCLC who developed diseases associated with disorders of the coagulation-fibrinolysis system under the treatment with ICI monotherapy.

Recent studies have shown that the association of early irAEs with the efficacy of ICIs in NSCLC, suggesting the relationship between systemic immune activation and the efficacy of ICIs [45,46]. The onset of common irAEs—such as rash, pyrexia, and endocrinopathies—have been reported to be early predictive factors of efficacy. Intriguingly, the patients who developed diseases associated with disorders of the coagulation-fibrinolysis system in association with the administration of ICIs tended to have better response to the therapy (the ORR was 72%) in view of the recent results from clinical studies, in which ORR to ICI monotherapy or ICI combination therapies was approximately 40–60% even in first-line setting [30,32,33]. In addition, the early onset of the coagulation-fibrinolysis system disorders were seen; the diseases associated with disorders of coagulation-fibrinolysis system developed within two cycles of ICI therapies for 67% patients and the median onset of cycle was one.

High PD-L1 expression on tumor cells mirrors immunologically ‘hot’ tumor, which are characterized by high infiltration of T cells, and the immune system in NSCLC patients with high tumor expression of PD-L1 are ready to be activated by ICIs [5,7,11]. High PD-L1 expression on tumor cells has been indeed associated with a high clinical response to ICIs in advanced NSCLC patients [32,33,47]. In addition, systemic immune activation by ICIs in peripheral blood of cancer patients have been confirmed in ICI responders [48,49]. In our study, all patients who developed diseases associated with disorders of coagulation-fibrinolysis system were positive for PD-L1, in addition, 82% of patients were strongly positive for PD-L1 on tumor (TPS ≥ 50%). Importantly, activated T cells promote procoagulant activity via induction of TF in monocytes/macrophages [16,50,51]. We demonstrated that T cell activation leads to abundant TF in PD-L1^high^ CD14^+^ monocytes. Therefore, an association between high PD-L1 expression on tumor cells, systemic immune activation by ICIs, the response to ICIs and disorders of the coagulation-fibrinolysis system during ICI therapy potentially exists in NSCLC patients who receiving immune checkpoint blockade.

### 4.2. Underlying Mechanisms of Disorders of the Coagulation-Fibrinolysis System as an irAE

The main irAEs of ICIs are skin, lungs, thyroid, endocrine, adrenal, pituitary, gastrointestinal tract, musculoskeletal, renal, and nervous system. Little is known about the risk of vascular events associated with ICIs [9,10]. The hypercoagulable state in cancer involves several complex interdependent mechanisms, including interaction among cancer cells, host immune cells, and coagulation-fibrinolysis system. Key roles in pathophysiology are played by TF, inflammatory cytokines, and platelets [16,17,18,19,25,35,52,53,54]. TF triggers the extrinsic coagulation cascade and cause disorders of coagulation-fibrinolysis system [24,25]. Importantly, monocytes have been shown to be the major source of intravascular TF in many diseases [16,25,27]. Therefore, we studied the impact of T cell activation on TF expression on monocytes in human PBMCs. In the current study, we demonstrated that T cell activation lead to monocyte activation and markedly increased PD-L1 on monocytes. We showed that PD-L1^high^ monocytes expressed higher TF compared to PD-L1^low^ monocytes, suggesting T cell activation by anti-PD-1/PD-L1 antibodies has the potential to induce high TF expression on peripheral PD-L1^+^ monocytes. 

The accumulating evidence suggests that activated T cells and APCs such as dendritic cells, macrophages, and monocytes are involved in provoking disorders of coagulation-fibrinolysis system [36,39,50,51,55]. Importantly, immune checkpoint blockade not only activates T cells but also activates APCs [37,56,57], indicating that anti-PD-1/PD-L1 monoclonal antibodies may trigger disorders of the coagulation-fibrinolysis system. Two possible mechanisms of the onset of diseases associated with disorders of the coagulation-fibrinolysis system under the treatment with ICIs are shown in Figure 6. 

There is a link between immune activation and thrombotic events within blood vessels [13,17,35,54,55,57]. TF is a primary initiator of fluid-phase blood coagulation and causes disorders of coagulation-fibrinolysis system [16,17,25]. Not only cancer cells but also activated monocytes and macrophages have been known to express abundant TF. It has been shown that monocytes/macrophages are a major source of circulating TF in the blood and their TF production is triggered by inflammatory cytokines such as IL-1β, TNF-α, and IFN-γ [17]. In addition, activated T cells have been shown to promote procoagulant activity via induction of TF in monocytes/macrophages [13,36,51]. Thus, the accumulating evidence suggests a potential risk of triggering disorders of coagulation-fibrinolysis system in association with ICI therapy within blood vessels; T cell activation by anti-PD-1/PD-L1 monoclonal antibody therapies may lead to promoting TF synthesis in monocytes/macrophages, which could result in triggering disorders of coagulation-fibrinolysis system such as DVT, PTE, and Trousseau’s syndrome in advanced cancer patients (Figure 6A).

In arteriosclerotic lesions, the activation of immune subsets including T cells and APCs—such as monocyte, macrophages, and dendritic cells—play critical roles in promoting plaque development, progression, and destabilization resulting in rupture and thrombus formation [14,39,50,52,53,58,59]. In addition, TF from activated immune subsets have been also suggested to be involved in the onset of ACS [35]. Therefore, the unwanted activation of immunity needs to be suppressed in atherosclerotic lesions. PD-1/PD-L1 signaling plays a critical role in inactivating immune cells and maintaining plaque stabilization in atherosclerotic lesions although T cell activation and pro-inflammatory cytokines such as interferon-γ (IFN-γ) and TNF-α are highly appreciated in terms of anti-tumor effects [14,39,50,58,60]. Activations of both CD4^+^ T cells and CD8^+^ T cells are suppressed by immune checkpoint pathways as a homeostatic mechanism in the atherosclerotic lesions, but, immune checkpoint blockade invigorates T-cell functions and activates APCs, also suggesting a potential risk of triggering disorders of coagulation-fibrinolysis system in arteriosclerotic lesions such as ACS and cerebral infarction in association with ICI therapy (Figure 6B).

Bleeding disorders are frequent in advanced cancer patients, and were observed in 2.7% of patients during a 1-year period [61]. Chemotherapeutic treatments have been known to heighten not only venous thromboembolism occurrence but also bleeding complications [17,62]. Although the underlying mechanism of bleeding disorders in cancer patients receiving ICI remains unclear, ICI-induced systemic immune activation followed by hypercoagulopathy and thromboembolic events consequently may cause a local consumptive coagulopathy, tissue damage, and endothelial injury could develop leading to internal and/or external bleeding complications such as purpura, intra-tumor hemorrhage, bronchial hemorrhage, and gastrointestinal hemorrhage.

Anticancer therapies have been shown to carry an increased risk of thrombotic events [19]. High rates of cancer-associated disorders of coagulation-fibrinolysis system have also been reported in cancer patients who are receiving antiangiogenic agents. In a meta-analysis of clinical trials of bevacizumab in combination with chemotherapy or interferon across a variety of cancers, the use of bevacizumab was associated with a 33% relative increase in the risk of venous thromboembolism [34]. However, the pathophysiology of anticancer therapy-associated disorders of coagulation-fibrinolysis system is not entirely understood [18,19,20]. Importantly, accumulating evidence from clinical and preclinical studies has shown that conventional and targeted anticancer agents have immunomodulatory or immune stimulatory effects and promote anti-tumor immunity [31,63,64,65,66], suggesting these immunomodulatory anticancer agents may also impact the coagulation-fibrinolysis system and increase the risk of thromboembolic/bleeding events through immune activation as well as ICIs.

### 4.3. Targeting PD-1/PD-L1 Signaling: A Double-Edged Sword in Cancer

A concept of “immune normalization” for the class of drugs called ICIs has recently been proposed [7]. Anti-PD-1/PD-L1 antibodies are monoclonal antibodies selectively targeting the PD-1/PD-L1 pathway and the mechanism of action of ICIs is thought to restore a lost antitumor immunity in the tumor microenvironment [7]. However, ICIs does not always change the immune balance to a favorable direction (Figure 7). ICIs selectively target the PD-1/PD-L1 pathway, however, do not selectively target the PD-1/PD-L1 signaling between tumor antigen-specific T cells and tumor cells, because immune cells expressing PD-1/PD-L1 not only exist in TME but also exist in peripheral blood and normal tissues, in addition, both PD-1 and PD-L1 are expressed on not only effector CD8^+^ T cells called “killer T cells”, but also a variety of immune cells including other T cell subsets, B cell subsets, and antigen-presenting cells including activated monocytes, macrophages, and dendritic cells [14,37,48,49,67]. Therefore, anti-PD-1/PD-L1 monoclonal antibodies can bind to various non tumor-specific T cells or non-tumor-directed immune subsets, which may lead to induce the unwanted activation of systemic immunity [59]. This may disturb the balance established between tolerance and autoimmunity and result in various irAEs.

PD-1 and PD-L1 are expressed in activated ‘non tumor-specific T cells’ as well as activated ‘tumor-specific T cells’. Thus, immune checkpoint blockade has a potential risk of shifting the systemic immune balance from tumor-specific T cell-mediated antitumor immune response to non-tumor-specific T cell-mediated immune response in cancer patients (Figure 6 and Figure 7A).

The crosstalk between APCs and T cells plays a key role in achieving efficient anti-tumor immune responses, which can be supported by various signals derived from T cells, such as IFN-γ [68,69,70,71,72]. The interaction provides crucial stimulatory signals for efficient expansion and development of effector functions to T cells (so-called “License to kill”) [70,71]. APCs including monocytes and macrophages express both PD-L1 and PD-1 (Figure 7B) [37,67]. IFN-γ from activated T cells not only activate APCs but also strongly induce PD-L1 expression on APCs to impede T cell function and maintain immune homeostasis, although IFN-γ is the most important cytokine implicated in antitumor immunity [68,73]. Therefore, blocking PD-1/PD-L1 signaling can activate APCs [67,73,74,75] and ICIs have a potential risk of shifting the immune balance from tumor-directed monocyte/macrophage activation to non-tumor-directed monocyte/macrophage activation through T cell activation, resulting in common and unexpected irAEs, such as disorders of coagulation-fibrinolysis system [13,14,15] (Figure 5, Figure 6 and Figure 7). In our current study, we demonstrated that T cell activation leads to monocyte activation and promotes the production of TF in PD-L1 high human peripheral CD14^+^ monocytes, suggesting that T cell activation followed by APC activation during ICI therapy may play a crucial role in triggering the diseases associated with disorders of the coagulation-fibrinolysis system.

### 4.4. Limitation

Our findings should be interpreted with caution in view of the limited samples, retrospective study, short observation period, and heterogeneity of study cohort (ICIs used in this study and prior lines of therapy); the results need to be confirmed in larger cohorts. Because the cancer-associated thromboembolic and bleeding events are common in advanced cancer patients and diverse asymptomatic disorders are present in single patients [18,19,20], it is conceivable that disorders of the coagulation-fibrinolysis system during ICI therapy could be unrecognized by clinicians and real incidence of the diseases associated with disorders of the coagulation-fibrinolysis system might be higher than that of our study. The cause-and-effect relationship between ICI and disorders of the coagulation-fibrinolysis system is not completely proven, although we showed T cell activation leads to promote production of TF in PD-L1^+^ monocytes in vitro. In our current study, we showed TF expression on monocytes, however, TF can also be induced in the endothelial cells of the vessel wall and smooth muscle cells under various pathologic conditions, and tumor cells also express abundant TF. Thus, various mechanisms of TF production should be considered in cancer patients receiving ICI therapy. Inflammatory cytokines derived from activated immune subsets by ICI have the potential to play a key role in pathophysiology of disorders of the coagulation-fibrinolysis system. Excessive inflammatory cytokines may induce tissue damage and endothelial injury, which could lead to TF production from various tissues. TF release from tumor cells killed by activated T cells may also trigger disorders of the coagulation-fibrinolysis system. Further studies including monitoring TF expression on circulating monocytes in cancer patients receiving ICI monotherapy are needed to unveil the mechanism of thromboembolic and bleeding complications in cancer immunotherapy. 

## 5. Conclusions

This is the first evidence suggesting the association between disorders of the coagulation-fibrinolysis system and immune activation by ICIs in cancer patients with PD-L1^+^ tumor. The present study may contribute to our understanding of the mechanism of disorders of the coagulation-fibrinolysis system in cancer patients and provide new insights into the complex interplay among cancer, host-immunity, and immunotherapy.

## Figures and Tables

**Figure 1 jcm-08-00762-f001:**
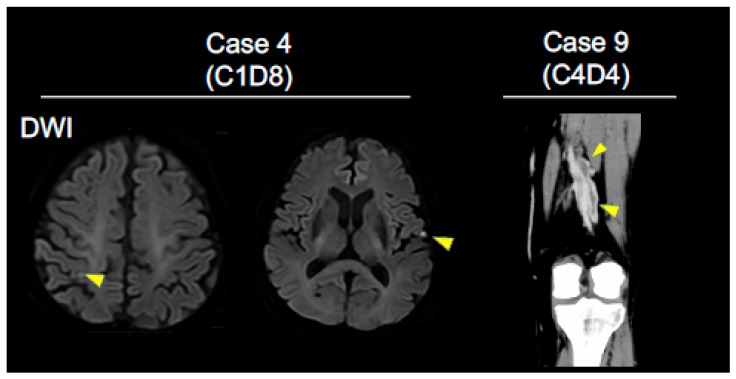
Key imaging in NSCLC patients who developed diseases associated with disorders of the coagulation-fibrinolysis. Brain MRI images (DWI) show multiple cerebral infarctions (arrowheads) (Case 4; cycle 1 day 8, C1D8, left panel). Contrast-enhanced CT image of leg vein shows deep vein thrombosis (arrowheads) (Case 9; C4D4, right panel).

**Figure 2 jcm-08-00762-f002:**
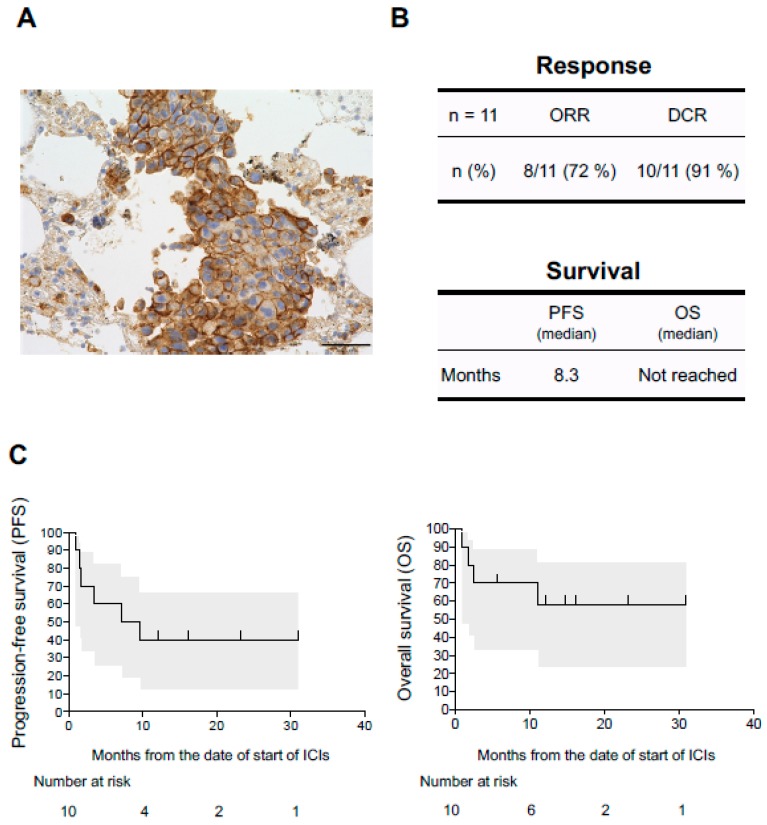
Association between efficacy of ICIs and the diseases associated with disorders of the coagulation-fibrinolysis system triggered by ICIs. (**A**) Representative immunohistochemical staining of the primary lung tumor in case 5, positive staining for programmed cell death-ligand 1 (PD-L1) (clone 22C3 pharmDx kit, tumor promotion score ≥75%). Scale bar 50 μm. (**B**) Upper panel shows objective response rate (ORR) and disease control rate (DCR) in NSCLC patients who developed diseases associated with disorders of the coagulation-fibrinolysis. Lower panel shows median PFS and median OS. (**C**) Kaplan–Meier curves are shown for PFS and OS in those patients. Gray areas indicate 95% confidence intervals of survival.

**Figure 3 jcm-08-00762-f003:**
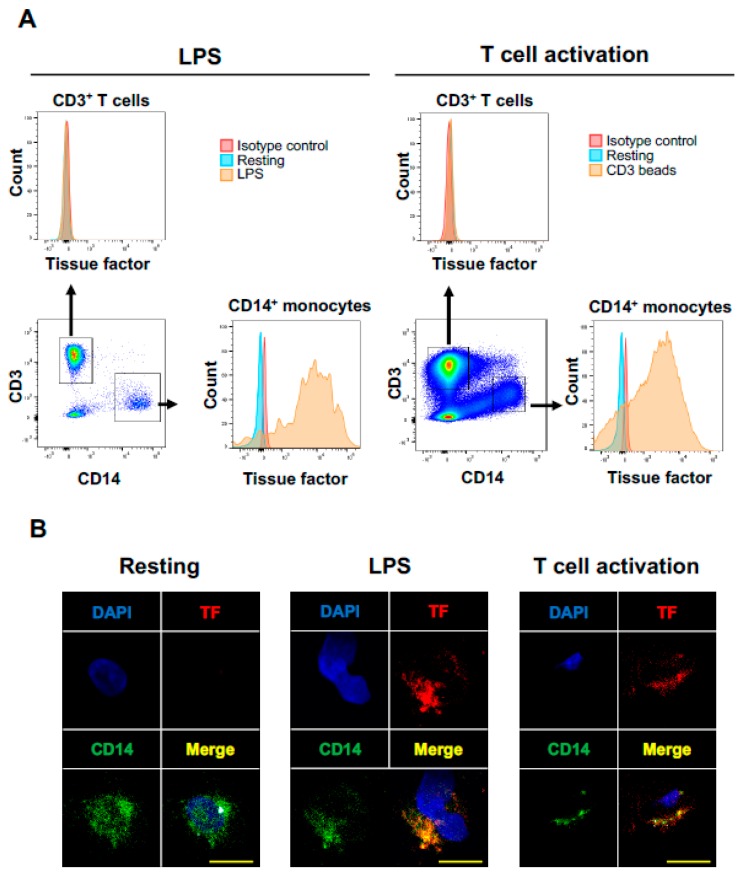
T cell activation induces TF expression on CD14^+^ monocytes. (**A**) Fresh PBMCs from healthy volunteers were cultured with lipopolysaccharide (LPS) to activate monocytes (left panels) or CD3/CD28/CD2 beads to activate CD3^+^ T cells (right panels). Gating strategy for analysis of CD3^+^ T cells or CD14^+^ monocytes is shown at each left lower panel. TF expressions were evaluated for CD3^+^ T cells and CD14^+^ monocytes. Representative histograms of TF on each population were shown. (**B**) Representative immunofluorescence images of PBMCs. Fresh PBMCs (left panels), PBMCs cultured with LPS (middle panels) or cultured with CD3/CD28/CD2 beads (right panels) were stained with antibodies against CD14 (green) and tissue factor (red). Nuclei were stained with DAPI (blue). Scale bars 10 μm.

**Figure 4 jcm-08-00762-f004:**
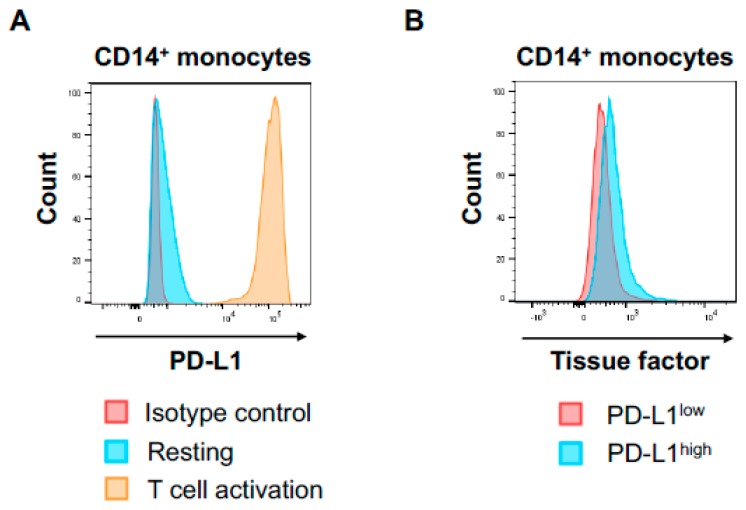
T cell activation increased PD-L1 on monocytes and PD-L1^high^ monocytes expressed higher TF compared with PD-L1^low^ monocytes. (**A**) Histogram of PD-L1 expression in CD14^+^ monocytes under resting condition or cultured with CD3/CD28/CD2 beads. (**B**) Histogram of TF in PD-L1^high^ and PD-L1^low^ CD14^+^ monocytes cultured with CD3/CD28/CD2 beads.

**Figure 5 jcm-08-00762-f005:**
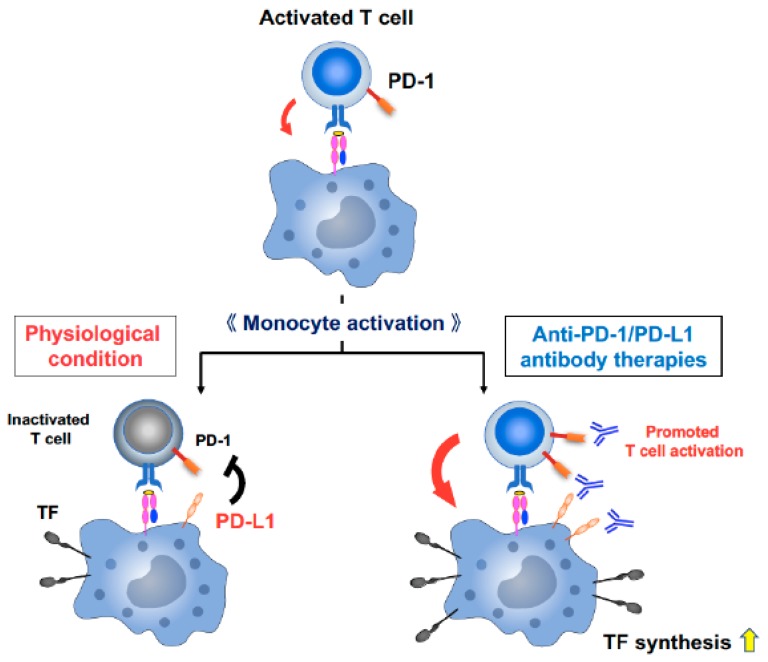
A model demonstrating the association between T cell activation by ICIs and TF production on APCs. T cell activation and proliferation are initiated through antigen recognition by the T cell receptor. The crosstalk between APCs and T cells provides crucial stimulatory signals for efficient expansion and development of effector functions to T cells and also induce activation of APCs and TF production. In physiological condition, upregulated PD-L1 on activated APCs suppresses T cells and results in end of immune activation as a homeostatic mechanism. However, ICIs provide a forced activation of T cells by blocking immune checkpoint pathways, which may lead to promoting further APC activation and abundant TF production, and may trigger disorders of the coagulation-fibrinolysis system.

**Figure 6 jcm-08-00762-f006:**
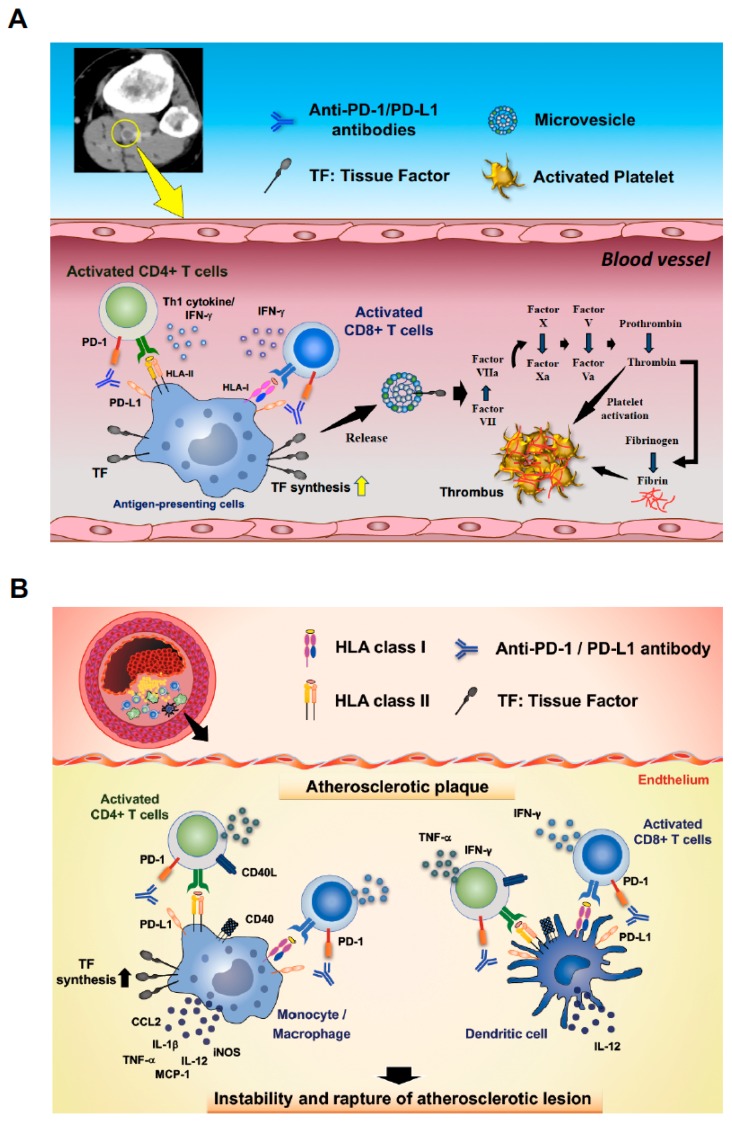
Hypothetical mechanisms of the onset of disorders of coagulation-fibrinolysis triggered by ICIs based on the current study and references. (**A**) The crosstalk between APCs and T cells provides crucial stimulatory signals for efficient expansion and development of effector functions to T cells. ICIs promote T cell activation by blocking the interaction between PD-1 and PD-L1 and induces IFN-γ and Th1 cytokine production, which play a crucial role in anti-tumoral effects. In turn, the IFN-γ and Th1 cytokines promote APC activation and TF synthesis in monocytes/macrophages. TF-containing membrane fragments or microvesicles released by the monocytes/macrophages could be a cause of distant thromboembolic events in advancer cancer patients. Microvesicles carrying TF activate factor VII. Conversion of factor VII to its active form (VIIa) in complex with TF triggers the production of other coagulation-related proteases in the coagulation cascade. The complex TF-factor VIIa converts factor X to activated factor X (factor Xa). Factor Xa with its cofactor, activated factor V (factor Va), activates prothrombin and generate thrombin, which is required to transform fibrinogen into fibrin and to activate platelets. This hypothetical mechanism could play a role in intravascular thrombosis in cancer patients receiving ICIs. A contrast-enhanced CT image of leg vein from the case 9 is shown on the right upper side. (**B**) In atherosclerotic lesions, T cells and APCs have been shown to be involved in promoting plaque development, progression, and destabilization of atherosclerotic lesions. Activated APCs, such as monocytes/macrophages and DCs, promote inflammation by secretion of pro-inflammatory mediators such as IL-12 and TNF-α or by promoting T cell activation. TF expression on APCs in atherosclerotic lesions is also promoted. Activated T cells produce pro-atherogenic cytokines such as IFN-γ and TNF-α that contribute to both the growth and destabilization of atherosclerotic lesions, which could result in rupture of the lesion. In contrast to cancer where T-cell activation and pro-inflammatory cytokines produced by immune subsets are highly appreciated, the unwanted activation of immune subsets needs to be suppressed in atherosclerotic lesions. As a homeostatic mechanism, activation of T cell subsets is suppressed by immune checkpoint pathways in the atherosclerotic lesions; however, ICIs could promote T-cell activation by blocking the immune checkpoint pathways. This evidence raises a hypothesis that ICIs might be involved in provoking the growth and destabilization of atherosclerotic lesions and causing disorders of coagulation-fibrinolysis by activation and recruitment of T cells and APCs in the atherosclerotic lesions.

**Figure 7 jcm-08-00762-f007:**
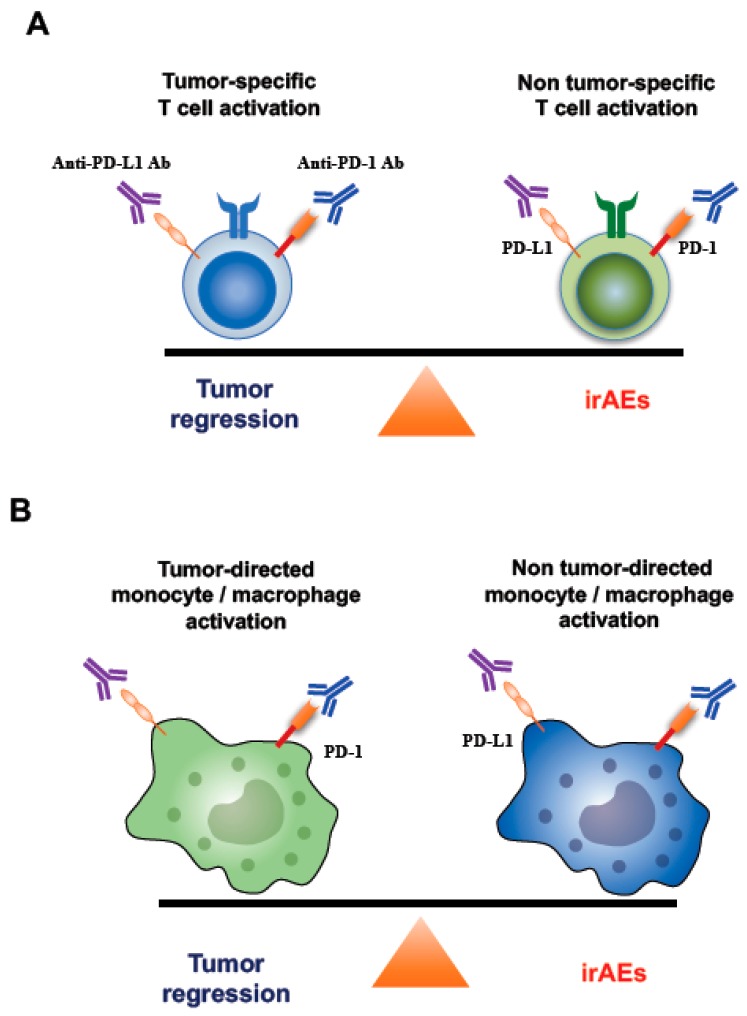
Underlying mechanisms of irAEs caused by activated T cells and monocytes/macrophages. (**A**) A model of immune balance between tumor-specific and non-tumor-specific T cells. ICIs can activate not only tumor-specific T cells but also non-tumor-specific T cells. Thus, ICIs have the potential to modulate the balance between tumor-specific T cell response and non-tumor-specific T cell response. PD-1/PD-L1 express on both tumor-specific and non-tumor-specific T cells. If non tumor-specific T cells are dominantly activated by ICIs, this may lead to the onset of irAEs. (**B**) A model of immune balance between tumor-directed monocyte/macrophage activation and non-tumor-directed monocyte/macrophage activation. PD-1/PD-L1 express on both tumor-directed and non-tumor-directed monocytes/macrophages. Thus, ICIs could activate both tumor-directed and non-tumor-directed monocytes/macrophages and have the potential to modulate the immune balance. If non tumor-directed monocytes/macrophages are dominantly activated by ICIs, this may lead to the onset of irAEs such as disorders of the coagulation-fibrinolysis system.

**Table 1 jcm-08-00762-t001:** Clinical features of NSCLC patients who developed diseases associated with disorders of coagulation-fibrinolysis system during immune checkpoint blockade therapy.

Patient	Age/Gender	Coagulation-fibrinolysis System AE	Hemorrhagic Complication	CTCAE Grade	Therapeutic Antibody/Treatment Line	Onset (Cycle)	Histopathology/Clinical Stage	PD-L1 Expression	Tumor Response	Reference
1	60/F	ACS		3	Nivolumab/2	2	Adeno/IVB	NE	PR	Ferreira et al. [22]
2	61/M	ACS, Cerebral lacunar infarction		3, 1	Nivolumab/3	11	Adeno/IVB	≥75%	CR	Tomita et al. [14]
3	63/M	Multiple cerebral infarcts, Intracranial hemorrhage	+	3, 5	Pembrolizumab/1	1	Adeno/IVB	≥75%	SD	Horio et al., [13]
4	72/M	Multiple cerebral infarcts		1	Atezolizumab/3	1	Sq/IIIA recurrence	≥50%	NE	
5	71/M	Gastrointestinal bleeding, Multiple cerebral infarcts	+	2, 1	Pembrolizumab/1	1, 6	Adeno/IVB	≥75%	CR	
6	67/M	Cerebral microbleed and lacunar infarction	+	1	Pembrolizumab/1	12	Sq/IVB	80%	PR	
7	48/F	DVT, PTE		2, 3	Pembrolizumab/1	1	Adeno/IVB	≥90%	PR	Kunimasa et al. [29]
8	76/M	Purpura, DVT	+	2, 2	Nivolumab/4	1, 2	Adeno (EGFRm)/IVB	20–30%	PD	
9	68/F	DVT, Bronchial hemorrhage	+	2, 2	Nivolumab/3	1, 4	Adeno/IVB	1–24%	PR	
10	81/M	Bronchial hemorrhage	+	2	Nivolumab/3	16	Sq/IVB	≥75%	PR	
11	74/M	Purpura	+	2	Pembrolizumab/1	4	Adeno/IVB	80%	PR	
12	79/M	Gastrointestinal bleeding	+	2	Pembrolizumab/1	1	Sq/IIIB recurrence	50–75%	SD	

Abbreviations: ACS, acute coronary syndrome; AE, adverse event; Adeno, adenocarcinoma; DVT, deep venous thrombosis; Sq, squamous cell carcinoma; SD, stable disease; PD, progression disease; PR, partial response; PTE, pulmonary thromboembolism; CR, complete response; M, male; F, female; NE, not examend. A newly developed purpura involving more than 10% of the body surface area (grade ≥ 2) was considered as disorders of coagulation-fibrinolysis system triggered by immune checkpoint blockade therapy in this study.

**Table 2 jcm-08-00762-t002:** Past medical history of NSCLC patients who developed diseases associated with disorders of coagulation-fibrinolysis system during immune checkpoint blockade therapy.

Patient	Age/Gender	Coagulation-Fibrinolysis System AE during ICI Therapy	Hemorrhagic Complication	Use of Anticoagulants or Antiplatelet Agents	Past Medical History of Diseases Associated with Disorders of Coagulation-Fibrinolysis System	Reference
1	60/F	ACS		none	none	Ferreira et al. [22]
2	61/M	ACS, Cerebral lacunar infarction		none	none	Tomita et al. [14]
3	63/M	Multiple cerebral infarcts, Intracranial hemorrhage	+	none	none	Horio et al. [13]
4	72/M	Multiple cerebral infarcts		none	none	
5	71/M	Gastrointestinal bleeding, Multiple cerebral infarcts	+	Aspirin, Clopidogrel	Coronary stenosis	
6	67/M	Cerebral microbleed and lacunar infarction	+	none	Cerebral bleed	
7	48/F	DVT, PTE		none	none	Kunimasa et al. [29]
8	76/M	Purpura, DVT	+	none	none	
9	68/F	DVT, Bronchial hemorrhage	+	none	none	
10	81/M	Bronchial hemorrhage	+	Aspirin, Dipyridamole	Myocardial infarction	
11	74/M	Purpura	+	none	none	
12	79/M	Gastrointestinal bleeding	+	Cilostazol, Clopidogrel	Cerebral infarction	

Abbreviations: ACS, acute coronary syndrome; AE, adverse event; DVT, deep venous thrombosis; PTE, pulmonary thromboembolism. No patients have been taking anticoagulants.

**Table 3 jcm-08-00762-t003:** AEs associated with disorder of coagulation-fibrinolysis system reported in clinical trials of ICIs in patients with advanced NSCLC.

Study [Reference]	No. of Patients Who Received ICI	Phase	Histology	Treatment	AEs Associated with Disorders of Coagulation-Fibrinolysis System	No. of Patients (%)
CheckMate017 [40]	135	III	Sq	Nivolumab	Not reported *	
CheckMate057 [44]	292	III	NonSq NSCLC	Nivolumab	Hemoptysis	16 (6%)
					Pulmonary embolism	1 (<1%)
					Cerebrovascular accident	1 (<1%)
KEYNOTE010 [41]	690	II/III	NSCLC	Pembrolizumab	Not reported *	
KEYNOTE024 [33]	154	III	NSCLC	Pembrolizumab	Not reported	
OAK [43]	425	III	NSCLC	Atezolizumab	Not reported *	
POPLAR [42]	144	II	NSCLC	Atezolizumab	Not reported *	

* Adverse events (AEs) occurred in less than 5% (CheckMate 017), 1% (KEYNOTE-010), 10% (OAK), or 10% (POPLAR) of patients who received immune checkpoint inhibitor (ICI) were not reported. Sq, squamous cell carcinoma.

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
