# Peer review of "Disorder of Coagulation-Fibrinolysis System: An Emerging Toxicity of Anti-PD-1/PD-L1 Monoclonal Antibodies"

_jcm, 2019, doi:10.3390/jcm8060762_

Reviewer 1 Report

The authors have retrospectively examined the charts of patients treated with checkpoint inhibitors at their institution for coagulation imbalances. This is based on the established understanding that the immune and coagulation systems are intricately connected. In addition, the authors have included an in vitro aspect to support their clinical hypothesis.

Major issues:

The authors should consider if there an appropriate control group for their case-series. Perhaps historical controls could help readers appreciate to what degree the coagulation abnormalities are due to treatment with the checkpoint inhibitors? As written, the authors are at-risk for over-interpreting their data.

Because DIC is the end-stage of virtually all abnormalities of coagulation, it seems inappropriate to exclude this adverse event. If the authors decide to keep this important exclusion, then the authors should further justify why DIC is an appropriate exclusion for a study focused on coagulation abnormalities.

This study is focused on coagulation abnormalities; however, the authors don’t seem to have an understanding of the patient’s coagulation-relevant history. I appreciate that a lot will be unknown, due to this being a retrospective study, but the author’s should make an appropriate attempt. This further helps readers interpret the adverse events data.

D-dimer just indicates that thrombin, factor XIII, and plasmin have worked, in-turn. Furthermore, it may take a prolonged time for the body to clear d-dimers. Without case-controls, this rise in d-dimers seems uninterpretable. For the author’s to suggest trending d-dimers based on this case-series seems to be an over-interpretation of their findings.

Suggestion: The authors may want to publish their in vitro data separately and first, then reference that work in their clinical work. This would enable the ideal readership to see relevant work.

Author Response

Response to the Reviewers’ comments

We thank the Reviewers for the time they have taken to go over our manuscript and for their helpful suggestions. Specific responses to the Reviewers’ comments follow below.

Responses to the comments from Reviewer #1

The authors have retrospectively examined the charts of patients treated with checkpoint inhibitors at their institution for coagulation imbalances. This is based on the established understanding that the immune and coagulation systems are intricately connected. In addition, the authors have included an in vitro aspect to support their clinical hypothesis.

Comment 1; The authors should consider if there an appropriate control group for their case-series. Perhaps historical controls could help readers appreciate to what degree the coagulation abnormalities are due to treatment with the checkpoint inhibitors? As written, the authors are at-risk for over-interpreting their data.

We thank the Reviewer for this comment. According to the Reviewer’s suggestion, we have incorporated the following sentences and Table 3 in the Discussion section of the revised main text in page 13, which shows the incidence of adverse events (AEs) associated with disorder of coagulation-fibrinolysis system in landmark clinical trials of ICIs in patients with advanced NSCLC. We have also added relevant references in revised main text.

The incorporated sentences;

    In most phase II/III clinical studies of ICIs, only AEs which were considered by the investigators to be related to the study therapy or high-incidence AEs (≥ 5-10% of patients who received ICI) have been reported (Table 3). Thus, not all AEs were known. AEs associated with the disorders of coagulation-fibrinolysis system have not been reported in five landmark clinical studies of ICIs in patients with advanced NSCLC, whereas hemoptysis (n=16, 6%), pulmonary embolism (n=1,<1%) and cerebrovascular accident (n=1, <1%) were reported in CheckMate057. However, CheckMate057 have not shown that the association of coagulation-fibrinolysis system disorder with the efficacy of ICIs.

Comment 2; Because DIC is the end-stage of virtually all abnormalities of coagulation, it seems inappropriate to exclude this adverse event. If the authors decide to keep this important exclusion, then the authors should further justify why DIC is an appropriate exclusion for a study focused on coagulation abnormalities.

We agree with the reviewer’s comment. According to Reviewer #1’s suggestion, we have removed the following sentence from the METHOD and Result section and also revised the relevant sentences in revised main text.

A removed sentence from method section of the main text in page 3:

[or disseminated intravascular clotting (DIC) (to exclude disease progression-associated disorder of coagulation-fibrinolysis system)]

Among 83 NSCLC patients, we had two patients who developed DIC during ICI therapy. These two patients had apparent pneumonia and sepsis with a procalcitonin increase during treatment with ICI, which followed by the onset of DIC. We have incorporated the following sentences in the Result section of the revised main text in page 4 to explain the reason why the patients who developed DIC were not included in Table 1.

    Disseminated intravascular coagulation (DIC) caused by pneumonia and sepsis accompanied with elevations of procalcitonin in blood were seen in two patients during treatment with ICI. However, ICI-related DIC without infectious diseases was not observed in current study. Thus, the two patients who developed DIC were not considered as coagulation-fibrinolysis system disorders triggered by ICI.

Comment 3; This study is focused on coagulation abnormalities; however, the authors don’t seem to have an understanding of the patient’s coagulation-relevant history. I appreciate that a lot will be unknown, due to this being a retrospective study, but the author’s should make an appropriate attempt. This further helps readers interpret the adverse events data.

We thank the Reviewer for this comment. According to the Reviewer’s suggestion, we have incorporated the following sentences and Table 2 in the Result section of the revised main text in page 7.

    Among twelve patients, four patients had past medical history of diseases associated with disorders of coagulation-fibrinolysis system. Three of the 4 patients had been taking antiplatelet agents (Table 2).

Comment 4;  D-dimer just indicates that thrombin, factor XIII, and plasmin have worked, in-turn. Furthermore, it may take a prolonged time for the body to clear d-dimers. Without case-controls, this rise in d-dimers seems uninterpretable. For the author’s to suggest trending d-dimers based on this case-series seems to be an over-interpretation of their findings.

Response to this comment: We agree with the reviewer’s comment. According to Reviewer #1’s suggestion, we have removed Figure 1B and 1C, and we have also removed relevant sentences in Result section of the revised main text in pages 7.  

We have revised the relevant figure legends for Figure 1B-C.

Comment 5;  Suggestion: The authors may want to publish their in vitro data separately and first, then reference that work in their clinical work. This would enable the ideal readership to see relevant work.

We thank the Reviewer for this helpful suggestion. However, we believe that our in vitro data which demonstrating T cell activation leads to promote production of TF in PD-L1high human peripheral CD14+ monocytes, are important to support our hypothesis of underlying mechanisms of disorders of coagulation-fibrinolysis system triggered by anti-PD-1/PD-L1 antibody therapy. Thus, we want to keep in vitro data in this manuscript. To make it clearer the reason why we focused on TF on monocytes in this study and why we conducted in vitro assay using human PBMCs, we have incorporated the following sentences and appropriate references in the Introduction and Discussion sections of the revised main text in pages 2 and 14.

Introduction section in page 2:

Tissue factor (TF) is a transmembrane cell surface glycoprotein that triggers the extrinsic coagulation cascade and is essential for hemostasis. TF binds the coagulation serine protease factor VII/VIIa (FVII/VIIa) to form a bimolecular complex that functions as the primary initiator of coagulation in vivo. Studies have shown that levels of circulating TF in the form of microparticles are increased in various diseases, including cardiovascular disease, sepsis, and cancer. In addition, circulating TF in blood has been suggested to be a cause of distant thromboses and contributes to the increased incidence of thrombosis observed in these diseases. Importantly, monocytes have been shown to be the major source of intravascular TF in many diseases. Therefore, we focused on the relationship between T cell activation and induction of TF expression on monocytes in peripheral blood mononuclear cells (PBMCs) in this study. 

Discussion section in page 14:

TF triggers the extrinsic coagulation cascade and cause disorders of coagulation-fibrinolysis system. Importantly, monocytes have been shown to be the major source of intravascular TF in many diseases. Therefore, we studied the impact of T cell activation on TF expression on monocytes in human PBMCs. In current study, we demonstrated that T cell activation lead to monocyte activation and markedly increased PD-L1 on monocytes. We showed that PD-L1high monocytes expressed higher TF compared to PD-L1low monocytes, suggesting T cell activation by anti-PD-1/PD-L1 antibodies has the potential to induce high TF expression on peripheral PD-L1+ monocytes.

Reviewer 2 Report

Globally, very nice work. 

Some remarks and questions:

You describe the incidence of thrombosis/bleeding events in pts with NSCLC treated with immune checkpoint inhibitors. You hypothesize that treatment with ICI may have caused these problems. 

- Most of the patients in your sample have an advanced age. Did you check complications, comorbidities as well as use of other drugs (anticoagulants or antiaggregants, other cancer therapies (bevacizumab etc.))? Did these patients already have cardiovascular disease or a history of thrombotic/hemorrhagic events? I think it's very important to at least check this as these could be the cause of the events.

- Are there other data on the hematologic and coagulation tests (PT, APTT, fibrinogeen, etc.) than D-dimeres? Do these support the hypothesis that coagulation can be affected by ICI-therapy?

- You describe the tissue factor pathway as a possible cause for thrombotic events. Why did you investigate this particular pathway in the context of ICI-therapy? Are there any other pathophysiologic mechanisms?

- Why do you suggest an association between immune activation by ICI and onset of disorders of coagulation/fibrinolysis from the ORR? The ORR in these 12 pts may be explained by the high PD-L1 score. 

- Are you preparing any clinical work or confirmation in a larger patient cohort?

Author Response

Response to the Reviewers’ comments

We thank the Reviewers for the time they have taken to go over our manuscript and for their helpful suggestions. Specific responses to the Reviewers’ comments follow below.

Responses to the comments from Reviewer #2

Globally, very nice work. Some remarks and questions:  You describe the incidence of thrombosis/bleeding events in pts with NSCLC treated with immune checkpoint inhibitors. You hypothesize that treatment with ICI may have caused these problems. 

Comment - Most of the patients in your sample have an advanced age. Did you check complications, comorbidities as well as use of other drugs (anticoagulants or antiaggregants, other cancer therapies (bevacizumab etc.))? Did these patients already have cardiovascular disease or a history of thrombotic/hemorrhagic events? I think it's very important to at least check this as these could be the cause of the events.

We thank the Reviewer for this comment. According to the Reviewer’s suggestion, we have incorporated the following sentences and Table 2 in the Result section of the revised main text in page 7.

    Among twelve patients, four patients had past medical history of diseases associated with disorders of coagulation-fibrinolysis system. Three of the 4 patients had been taking antiplatelet agents (Table 2). Antiangiogenic agents including bevacizumab have not been used prior to ICI in these 12 patients.

Comment ; - Are there other data on the hematologic and coagulation tests (PT, APTT, fibrinogeen, etc.) than D-dimeres? Do these support the hypothesis that coagulation can be affected by ICI-therapy?

We thank the Reviewer for this comment. PT and APTT did not show changes in patients who developed disorder of coagulation-fibrinolysis system. Unfortunately, we did not have data for Fibrinogen in most patients. In a few patients, D-dimer was monitored at pre- and post ICI therapy and also at pre- and post onset of diseases associated with disorder of coagulation-fibrinolysis system. D-dimer is a product of fibrin degradation and have been proposed as a promising predictor for postoperative complications. Thus, we thought that our data are meaningful to readership and the increases of D-dimer followed by the onset of disorders of coagulation-fibrinolysis system after ICI therapy may support our hypothesis.  However, the reviewer #1 pointed out the risk of an over-interpretation of our D-dimer data. Therefore, we decided to remove D-dimer data and relevant sentences from our manuscript.

Comment 3-1; - You describe the tissue factor pathway as a possible cause for thrombotic events. Why did you investigate this particular pathway in the context of ICI-therapy?

We thank the Reviewer for this helpful suggestion. To make it clearer the reason why we focused on TF on monocytes in this study and conducted in vitro assay using human PBMCs, we have incorporated the following sentences and appropriate references in the Introduction and Discussion sections of the revised main text in pages 2 and 14.

Introduction section in page 2:

 Tissue factor (TF) is a transmembrane cell surface glycoprotein that triggers the extrinsic coagulation cascade and is essential for hemostasis. TF binds the coagulation serine protease factor VII/VIIa (FVII/VIIa) to form a bimolecular complex that functions as the primary initiator of coagulation in viv. Studies have shown that levels of circulating TF in the form of microparticles are increased in various diseases, including cardiovascular disease, sepsis, and cancer. In addition, circulating TF in blood has been suggested to be a cause of distant thromboses and contributes to the increased incidence of thrombosis observed in these diseases. Importantly, monocytes have been shown to be the major source of intravascular TF in many diseases. Therefore, we focused on the relationship between T cell activation and induction of TF expression on monocytes in peripheral blood mononuclear cells (PBMCs) in this study. 

Discussion section in page 14:

TF triggers the extrinsic coagulation cascade and cause disorders of coagulation-fibrinolysis system. Importantly, monocytes have been shown to be the major source of intravascular TF in many diseases. Therefore, we studied the impact of T cell activation on TF expression on monocytes in human PBMCs. In current study, we demonstrated that T cell activation lead to monocyte activation and markedly increased PD-L1 on monocytes. We showed that PD-L1high monocytes expressed higher TF compared to PD-L1low monocytes, suggesting T cell activation by anti-PD-1/PD-L1 antibodies has the potential to induce high TF expression on peripheral PD-L1+ monocytes.

Comment 3-2; - Are there any other pathophysiologic mechanisms?

We thank the Reviewer for this helpful suggestion. To show the other potential mechanisms, we have incorporated the following sentences in the Discussion sections of the revised main text in pages 19.

In current study, we showed TF expression on monocytes, however, TF can also be induced in the endothelial cells of the vessel wall and smooth muscle cells under various pathologic conditions, and tumor cells also express abundant TF. Thus, various mechanisms of TF production should be considered in cancer patients receiving ICI therapy. Inflammatory cytokines derived from activated immune subsets by ICI have the potential to play a key role in pathophysiology of disorders of coagulation-fibrinolysis system. The excessive inflammatory cytokines may induce tissue damage and endothelial injury, which could lead to TF production from various tissues. TF release from tumor cells killed by activated T cells may also trigger disorders of coagulation-fibrinolysis system.

Comment ; - Why do you suggest an association between immune activation by ICI and onset of disorders of coagulation/fibrinolysis from the ORR? The ORR in these 12 pts may be explained by the high PD-L1 score. 

We thank the Reviewer for this helpful suggestion. To make it clear that, we have revised relevant sentences in discussion section and have incorporated the following sentences and appropriate references in Discussion section of the revised main text in pages 13 and 14.

High PD-L1 expression on tumor cells mirrors immunologically “hot” tumor, which are characterized by high infiltration of T cells, and the immune system in NSCLC patients with high tumor expression of PD-L1 are ready to be activated by ICIs. High PD-L1 expression on tumor cells has been indeed associated with a high clinical response to ICIs in advanced NSCLC patients. In addition, systemic immune activation by ICIs in peripheral blood of cancer patients have been indeed confirmed in ICI responders. In our study, all patients who developed diseases associated with disorders of coagulation-fibrinolysis system were positive for PD-L1, in addition, 82% of patients were strongly positive for PD-L1 on tumor (TPS≥50%). Importantly, activated T cells promote procoagulant activity via induction of TF in monocytes/macrophages. We demonstrated that T cell activation leads to abundant TF in PD-L1high CD14+ monocytes. Therefore, an association between high PD-L1 expression on tumor cells, systemic immune activation by ICIs, the response to ICIs and disorders of coagulation-fibrinolysis system during ICI therapy potentially exists in NSCLC patients who receiving immune checkpoint blockade.

Comment ; - Are you preparing any clinical work or confirmation in a larger patient cohort?

We thank the Reviewer for this comment. As we have wrote in original manuscript; Further studies including monitoring TF expression on circulating monocytes in cancer patients receiving ICI monotherapy are needed, we are planning to analyze TF expression on NSCLC patients receiving ICI and correlate it with clinical response, PFS and OS.

Round  2

Reviewer 1 Report

I appreciate the responses and updates from the authors.